# Fabrication and Biological Analysis of Highly Porous PEEK Bionanocomposites Incorporated with Carbon and Hydroxyapatite Nanoparticles for Biological Applications

**DOI:** 10.3390/molecules25163572

**Published:** 2020-08-06

**Authors:** P. D. Swaminathan, Md. Nizam Uddin, P. Wooley, Ramazan Asmatulu

**Affiliations:** Department of Mechanical Engineering, Wichita State University, 1845 Fairmount, Wichita, KS 67260, USA; mxuddin7@shockers.wichita.edu (P.D.S.); engrnizam02@gmail.com (M.N.U.); isratja2016@gmail.com (P.W.)

**Keywords:** PEEK, hydroxyapatite, carbon particles, bionanocomposites, cytotoxicity, cell viability, bone marrow cell

## Abstract

Bone regeneration for replacing and repairing damaged and defective bones in the human body has attracted much attention over the last decade. In this research, highly porous polyetheretherketone (PEEK)/hydroxyapatite (HA) bionanocomposite scaffolds reinforced with carbon fiber (CF) and carbon nanotubes (CNTs) were fabricated, and their structural, mechanical, and biological properties were studied in detail. Salt porogen (200–500 µm size) leaching methods were adapted to produce porous PEEK structures with controlled pore size and distribution, facilitating greater cellular infiltration and biological integration of PEEK composites within patient tissue. In biological tests, nanocomposites proved to be non-toxic and have very good cell viability. In addition, bone marrow cell growth was observed, and PEEK/HA biocomposites with carbon particles showed increased cell attachment over the neat PEEK/HA composites. In cell viability tests, bionanocomposites with 0.5 wt% CNTs established good attachment of cells on disks compared to neat PEEK/HA biocomposites. A similar performance was seen in culture tests of bone marrow cells (osteoblasts and osteoclasts). The 0.5 wt% CF for osteoblasts and 1 wt% CNTs for osteoclasts showed higher cell attachment. The addition of carbon-based nanomaterials into PEEK/HA has been identified as an effective approach to improve cell attachment as well as mechanical and biological properties. With confirmed cell attachment and sustained viability and proliferation of the fabricated PEEK/HA/CNTs, CF bionanocomposites were confirmed to possess excellent biocompatibility and will have potential uses in bone scaffolding and other biomedical applications.

## 1. Introduction

Research on the use of nano-biomaterials for orthopedic applications has recently gained considerable attention. Bone replacements have been increasing due to a higher number of accidents, birth defects, and many diseases, such as bone infections, tumors resulting in bone fracture, and bone loss [1,2]. Bone, a complex porous composite with unique properties of remodeling, can adapt its microstructure to external mechanical stress. Hence, incorporating nanocomposites as bone scaffolds has great potential and could be a viable solution for these issues. Bone scaffolds must be highly porous and provide support to the skeleton. They serve as a template for bone regeneration and must biodegrade at the rate of bone growth [3,4,5]. Bioresorbable scaffolds, i.e., porous constructs, seeded with appropriate types of cells should provide a template for tissue regeneration, while slowly resorbing to leave no foreign substances in the body and thereby reducing the risk of inflammation. In the past few years, polymer/carbon-based composites have gained increasing interest in the field of tissue engineering [6,7,8,9,10,11].

Polyetheretherketone (PEEK) has become a very interesting biomaterial for scientists and a promising good alternative for medical implantation because of its outstanding combination of toughness, stiffness, thermo-oxidative stability, chemical and solvent resistance, flame retardancy, and retention of physical properties at high temperatures [12,13,14,15]. Although the incorporation of hydroxyapatite (HA) into PEEK ameliorates the biological activities of PEEK, the resulting composite becomes brittle and lacks load-bearing mechanical properties. Different types of nanomaterials and modification methods involving PEEK to resolve this drawback have been reported in the literature [16,17,18]. In a recent study, HA was incorporated into PEEK to fabricate PEEK/HA biocomposites using a compounding and injection-molding technique, and the mechanical and bioactivity of the composites—i.e., cell attachment, proliferation, spreading, and alkaline phosphatase (ALP) activity of MC3T3-E1 cells—and apatite formation after immersion in simulated body fluid (SBF) and osseointegration in a rabbit cranial defect model were investigated [19]. Test results reveal that the PEEK/HA composite exhibited better bioactivity than that of ultra-high molecular weight polyethylene (UHMWPE) and pure PEEK. After immersion in SBF for 7 days, apatite islands formed on the PEEK/HA composite, and bone contact and new bone formation around the PEEK/HA composite were more noticeable than those of the UHMWPE and pure PEEK. To improve the bioactivity of HA, the silane coupling agent KH560 (γ-(2,3-epoxypropoxy) propytrimethoxysilane) was used to fabricate PEEK/HA composites via a hot-press molding method [20]. At 5 wt% HA loading, the tensile strength of the PEEK/m-HA composite was maximal, and the growth of the bone tissues around the composite was greater than in the unmodified one. These results further support the application of the PEEK/HA composite in biomedical fields. Hybrid composite scaffolds composed of PEEK/HA/polyglycolicacid (PGA) were developed via selective laser sintering (SLS), and their biological properties were studied [21]. The hybrid scaffolds demonstrated good apatite-forming ability with increasing HA content after immersion in SBF. Moreover, the degree of cell attachment and proliferation was greater than in the PEEK/PGA scaffolds. This study also proved the potential of PEEK/HA/PGA scaffolds for tissue regeneration. In another study, strontium (Sr)-containing PEEK/HA composites were fabricated and the mechanical and biological properties studied. Experimental results showed that PEEK/HA/Sr composites possess excellent physical properties and biological activity [22].

Numerical tests were conducted to investigate the biomechanical characteristics of two types of cages, i.e., PEEK/HA/carbon fiber (CF)and titanium combined with internal pedicle screw fixation, in a lumbar model to provide experimental evidence for clinical application [17]. Test results indicated that the von Mises peak stresses of the bone graft of the PEEK/HA/CF group were at least 2.2 times those of the titanium group. The angular variation of both groups was similar and could increase the load transfer through the bone graft and promote bone fusion. The biocompatibility of PEEK/HA/CF composites was investigated by co-culturing them with osteoblasts in vitro [18]. The quantitative assessment for the cytotoxicity of the biomaterials was measured by the cell relative growth rate (RGR). The proliferation index of the co-cultured cells and ALP activity were measured to study the effect of PEEK/HA/CF composites. Results showed that the PEEK/HA/CF composites have no cytotoxicity to osteoblasts. After 7 days, the ALP activity was the highest on the surface of the PEEK/HA/CF composites, and the osteoblast cells co-cultured with the PEEK/HA/CF composites adhered well to the biomaterial. Experimental results suggest that PEEK/HA/CF composites have good biocompatibility in vitro and are a novel orthopedic implanted material.

However, extensive study is still necessary to clarify which method is more appropriate, and long-term clinical studies are imperative for medical implant materials and applications. To address scaffold biodegradability and biocompatibility issues, this study focuses on the fabrication, characterization, and biological properties of highly porous PEEK/HA bionanocomposites incorporating CNTs and CF nanomaterials. Melt-casting and salt porogen leaching methods were adopted for the fabrication of highly porous PEEK foams. The fabrication process was conducted in an inert atmosphere to prevent oxidation of carbon under the high melting temperature of PEEK. The novelty of the present work is that highly robust PEEK scaffolds incorporated with various bone growth promoters (HA and carbon particles) were produced using the salt porogen technique, which will greatly improve the mechanical and bioactivity of new bionanocomposites. This article is a continuation of work focusing on the biological study by Uddin et al. [23], in which the mechanical properties are described in the article “Mechanical properties of highly porous PEEK bionanocomposites incorporated with carbon and hydroxyapatite nanoparticles for scaffold applications” [23]. The fundamental knowledge and skills gained through this study can be useful for advancing the properties of functional scaffolds to address some of the problems in this field.

## 2. Results and Discussion

### 2.1. Mechanical Properties

Compression tests were conducted on the fabricated bionanocomposite scaffolds. Compression test results of the fabricated bionanocomposites show that, compared to the neat PEEK scaffold, the addition of HA, CF, and CNTs significantly enhanced the modulus and yield strength of these biomaterials [23]. As observed in the compression tests, the incorporation of 0.5 wt% CNTs into PEEK/HA resulted in greater mechanical properties than all other compositions studied in this work. The modulus of elasticity and yield strength of PEEK/HA/CNT (0.5 wt%) bionanocomposites with 75% porosity provided 252.91 MPa and 4.51 MPa, as compared to neat PEEK of 66.46 MPa and 1.98 MPa, respectively [23]. In contrast, incorporating a higher amount of carbon particles into PEEK reduced the mechanical properties. This may be due to the agglomeration of the carbon particles and lack of matrix support, which act as stress raisers in the structure. In addition, the functionalization of the CF and CNTs resulted in uniform dispersion via covalent bonding and provided higher mechanical strength [14,24]. The mechanical properties of cortical and cancellous bones have been reported by Michael et al. [25]. However, cortical bone with 5%–30% porosity is denser than cancellous bone with 70%–95% porosity. Cancellous bone has a strength of 0.1–30 MPa, elasticity of 0.02–0.5 GPA, and strain of 5%–7%. The mechanical properties, i.e., compression modulus, yield strength, and elongation at break, of the bionanocomposites studied here fall within the range of cancellous bone.

### 2.2. Porosity of Bionanocomposites and Structural Analysis

A porosity of 75% or higher is necessary to maintain a space that is the right shape and size for tissue formation. For the best vascularization, a minimum pore size of 100 µm and porosity of up to 90% are necessary [26]. In this study, the pore size and interconnectivity of the nanocomposite are in the order and within the designed sizes and shapes, usually varying between 240 and 310 μm, as obtained from the micro-computed tomography test [23]. Neat PEEK (75% porosity) appears as a spongy foam compared to PEEK/HA composites. Moreover, the addition of HA improved the interconnectivity of the PEEK foams. The PEEK/HA nanocomposite (85% porosity) with CNT has better interconnectivity compared to CF, which may be due to the higher volume of nanoparticles of CNT compared to the micron particles of CF. In addition, PEEK/HA nanocomposites (85% porosity) with CF exhibit edges of loosely connected cells.

Representative scanning electron microscopy (SEM) images of the fractured surfaces of the bionanocomposites are shown in Figure 1. It can be observed from Figure 1a that the polymer matrix is quite ductile, as characterized by the presence of a rough surface. With the addition of HA, CNTs, and CF, these fillers can be readily observed in Figure 1b,c. It can be also observed that these nanoparticles are dispersed randomly in the polymer matrix. Some small holes from the pull-out of the CF were seen in the fracture cross-sections of the materials (Figure 1c). From these images, it can be concluded that some of the HA, CNT, and CF nanoparticles were embedded throughout the PEEK polymer and a highly porous structure is observed, which may be useful for scaffolding.

### 2.3. Biological Properties

The day 3 cell cultures with day 1 supernatant and various concentration ratios of sodium dodecyl sulfate (SDS) were studied. SDS does not allow cells to grow at higher concentrations; however, at 1:16 and 1:64 concentrations, cell growth is moderate. It was observed that the concentration of cells was significantly higher with lower additions of CNTs as compared to higher additions of CNTs in the media. In addition, the incorporation of HA nanoparticles ameliorates greater cell growth. Moreover, with increasing porosity, the rate of cell growth that takes place is higher, which is also like the neat PEEK foam. Microscopic images of day 3 cell cultures with day 1 supernatant in a 1:4 concentration ratio of bionanocomposites with 0.5 wt% CF and 1 wt% CNTs for 75% and 85% porosities are presented in Figure 2 and Figure 3.

The cytotoxicity and cell viability test results are illustrated in Figure 4. All bionanocomposites were non-toxic, and cell viability was mainly good. It is worth noting that nanomaterials may induce a cytotoxic effect on biological cells and the cytotoxicity of carbon materials is dependent on their dimensions. Moreover, the incorporation of CF in PEEK/HA nanocomposites exhibited a higher percentage of cell growth than the incorporation of CNTs. Compared to CNT addition, CF has been reported to have higher cell viability in the literature [27]. For encapsulating CF, the polymer matrix of nanocomposites is an effective material [28]. In addition, the cell viability of the nanocomposite reveals that HA enhances the cell viability—the greater the percentage of HA exhibited, the higher the cell viability [29]. Furthermore, CF and CNTs both increased the cell viability, except for the 2 wt% CNTs (primarily concentration effects). A dramatic increase in cell viability can be observed on PEEK/HA/CNT nanocomposites compared to the PEEK control. However, the bionanocomposites with the addition of 0.5 wt% CNTs and 1 wt% CF showed higher cell viability than all other samples, which proves that the addition of CNTs helps to attach more cells to the sample disk at lower concentrations of CNTs. Therefore, carbon materials embedded firmly in the polymer matrix of bionanocomposites and acted as excellent substrates for cells’ adhesion, growth, and cell viability.

### 2.4. Bone Marrow and Raw Cell Cultures on Bionanocomposites

Bone marrow and raw cell culture studies indicate that the introduction of CF and CNTs enhances cell attachment on the PEEK nanocomposites, compared to that of neat PEEK. Figure 5 shows the ALP measurement (OD405/µg protein/0.00889) from osteoblasts and undifferentiated bone marrow cells, demonstrating that test results for the PEEK disk with 1 wt% CNTs and 0.5 wt% CF composition (75% porosity) are closer to each other, and both indicate higher ALP expression than the neat PEEK.

The telomerase repeated amplification protocol (TRAP) expression (OD405/µg protein/0.00889) results of osteoclasts from raw cells are depicted in Figure 6. Even though the CNTs show slightly higher TRAP expression than the CF, the 0.5 wt% CF has higher expression. It can be concluded that both CNTs and CF help to improve cell differentiation on the PEEK nanocomposite. It is reported that the hybridization of HA filler with CF promotes osteoblastic adhesion and proliferation [28]. Moreover, CNTs and CF both enhance osteoblastic adhesion and differentiation by promoting protein–material interactions [30]. Elias et al. reported that osteoblast adhesion, proliferation, and ALP activity on CF increased with decreasing fiber diameter in the range of 60–200 nm [31]. In vitro cell culture confirmed that the cells attached, spread, and proliferated well on the fabricated bionanocomposites. Our results indicate that the bionanocomposite scaffolds possess excellent mechanical properties, good apatite-forming ability, and cytocompatibility and are potential future scaffolding and tissue engineering applications in many biomedical fields.

## 3. Materials and Methods

### 3.1. Materials

Dexamethasone, ascorbic acid, beta-glycerophosphate, and HA (size < 200 nm) were purchased from Sigma-Aldrich, Steinheim, Germany. Raw cells were collected from the American Type Culture Collection research center, Manassas, VA, USA. The polyacrylonitrile (PAN)-based carbon fiber, with a diameter of 7 µm and length of 80–100 µm, was supplied by E&L Enterprises, Inc. Galliano, LA, USA. The catalytic multi-walled CNTs used in this research had a diameter of 140 nm and a length of 7 µm and were purchased from MER Corporation, Tucson, AZ, USA. Pure ocean salt (NaCl), ranging in size from 200 to 500 μm, was supplied by SaltWorks^®^ (Seattle, WA, USA). The PEEK used for this research is Victrex, Lancashire, UK, PEEK 150P, high-performance thermoplastic material, unreinforced, semi-crystalline, coarse powder for extrusion compounding, easy flow, and natural color. The tensile strength, tensile modulus, and elongation at break of PEEK are 100 MPa, 3.7 GPa, and 15%, respectively. Table 1 summarizes the bionanocomposite samples’ codes, compositions, and the various tests performed in this study.

### 3.2. Fabrication of PEEK Bionanocomposite

#### 3.2.1. Functionalization of CF/CNTs

The functionalization of CF/CNTs with carboxylic acid supports interfacial bonding and increases uniform dispersion. In this study, nanoparticles and carboxylic acid were mixed in a ratio of 1:100 and stirred with a magnetic bar on a hot plate at an elevated temperature of 250 °C for 4 h at 500 rpm. Neutralization of the mixture was necessary to remove acid from the functionalized nanoparticles. To neutralize the functionalized nanoparticles, they were mixed with 750 mL of pure water, stirred again for 10 min, and then vacuum filtered. This cycle was repeated at least seven times to obtain effective neutralization. The neutralized and vacuumed-filtered nanoparticles were dried in an oven overnight at a temperature of 85 °C.

#### 3.2.2. Fabrication of PEEK Bionanocomposite

The required amounts of functionalized CF/CNTs (0.5, 1.0, and 2.0 wt%) with 20 mL of solvent (toluene) in a test tube were sonicated using a Sonics^®^ Vibra-Cell™, Model VCX 130 Newtown, CT, USA at 70% capacity for 15 min. Then, the required amounts of PEEK (15/25) and HA (20%) were added to the sonicated CF/CNTs, followed by the addition of 10 mL of solvent to make a slurry for effective mixing. This mixture was sonicated in four cycles each of 15 min, with a 5-min interval between each cycle. The solvent was drained from the sonicated slurry, allowed to dry for 48 h at room temperature, and then further dried at 120 °C in an oven for 3 h. The sonicated mixture was hand-ground in a mortar for 2 min to break up any solid particles and clusters. Then, 75%/85% salt porogen was added to the mixture of PEEK/HA/CF/CNTs and mixed uniformly using a Fisher Scientific (Waltham, MA, USA) pulsing vortex mixer (115VAC, 150 watts, 50/60 Hz, 1 phase for 10 min at 3000 rpm) to distribute the salt porogen evenly and attain proper pore size and properly interconnected pores.

Finally, the mixture was cast to obtain the PEEK bionanocomposites. Casting involved two steps: (a) preparation of the mold and (b) melting of the nanocomposite mixture in an appropriate atmosphere without oxidization. The mold, or die, used was a 5/8-in aluminum tube to obtain a smooth circular-shaped casting to facilitate machining, and the ethanol-cleaned mold was sprayed with a high-temperature release agent (Slide, Hi-Temp 1800, Wheeling, IL, USA) and allowed to dry for 15 min. The nanocomposite/salt porogen mixture was placed in the mold, mixed lightly with a stiff wire, hand-pressed with an aluminum rod the same size as the mold, and wrapped with thin aluminum foil. The nanocomposite and salt porogen mixture was melted using a Barnstead high-temperature Thermodyne 1300 furnace (Fort Wayne, IN, USA) in a regular atmosphere at 400 °C for 4 h and allowed to cool to room temperature within the furnace. The nanocomposite mixture was melted in an inert atmosphere using a Sentro Tech STT-1600-2.75-12 high-temperature vacuum tube furnace (Strongsville, OH, USA) to avoid oxidization from the presence of carbon at 400 °C for 4 h at a pre-fixed ramp-up and cooling rate (4 °C per min). Then, enough argon gas was pumped through the vacuum tube from a valve at one end and sealed. The casting was left to cool to room temperature within the furnace. The bionanocomposite fabrication process is illustrated in Figure 7.

#### 3.2.3. Leaching of Salt Porogen

The machined nanocomposite samples were leached out using clean water for 3 days, changing the water at regular intervals of 6–8 h. During the water change, samples were kept in an underwater flow for 10 min, which helped to remove any solid salt porogen particles, and then vacuumed for 15 min during every 24 h of leaching. The samples were dried overnight and further dried in a furnace at 110°C for 2 h to evaporate any water particles. 

### 3.3. Morphology, Cytotoxicity, and Cell Viability Tests

The morphology of the prepared nanocomposite was characterized by the SEM device (FEI Nova Nano SEM 450, Hillsboro, OR, USA). Using the SEM, several areas of the selected samples were imaged to inspect using the same magnifications. To study and comprehend the toxicity and cell viability, in vitro tests were carried out on the nanocomposites, including neat PEEK, PEEK/HA, and PEEK/HA with CF/CNTs, in various compositions and volumes of porosity. These samples were sterilized in an autoclave for two cycles at 121 °C of 30 min each. Then, the sample disks were transferred to two 24-well plates, and 750 μL of Dulbecco’s modified eagle medium (DMEM) was added to each well and incubated overnight at 37 °C. The day 1 supernatant was collected and stored at −20 °C. L929 cells were grown at 37 °C overnight in 96-well plates; the cell count was approximately 50,000 cells/100 μL/well for the cytotoxicity study. Day 1 supernatant was added to the wells at dilution rates of 1:1, 1:4, 1:16, and 1:64 and incubated overnight at 37 °C. The dilution steps were as follows: (i) 100 μL of (1:1) supernatant added to 300 μL of medium (1:4); (ii) 100 μL (step (a) supernatant) added to 300 μL of medium (1:16), and (iii) 100 μL (step (b) supernatant) added to 300 μL of medium (1:64). All four 200 μL concentrations were added to the wells. The seventh row of the column was filled with 10% of SDS of similar dilutions, the last row of well 96 was filled with 200 μL of the medium, and the plate was incubated for 3 days at 37 °C.

After the incubation period, the sample disks were placed into two new 24-well plates with 50,000 cells/100 μL/well, and 500 μL of the fresh medium was added and incubated overnight. The old medium was removed, and fresh medium with 80 μL of 3-(4, 5-dimethylthiazol-2-yl)-2, 5 diphenyltetrazolium bromide (yellow tetrazole) (MTT) was added and kept for 6 h. After 6 h, the medium was removed and transferred into the new 96-well plates, and 10% SDS was added. The disks were centrifuged to remove the medium and transferred to a second 96-well plate with 10% SDS, and the optical density (OD) was read at 590 nm to obtain the cell growth. The biological tests were repeated to check whether there were any inconsistencies. The quantitative data in this work were expressed as mean ± standard deviation and the statistical analysis of the data was performed by using one-way analysis of variance (ANOVA) and *p* < 0.05 was considered statistically significant.

### 3.4. Bone Marrow and Raw Cell Culture on Bionanocomposite Materials

Bone resorption and formation are the prime functions in bone remodeling. Osteoclast and osteoblast cells are responsible for these actions. Five different nanocomposites (PEEK/HA and PEEK/HA with 0.5 and 1 wt% of CF and CNTs) were autoclaved for 30 min at 121 °C, transferred into two 24-well plates with 0.5 mL of DMEM per well, and incubated overnight at 37 °C. Bone marrow cell and raw cell cultures were carried out to study and understand the trend of cell growth on porous composites. Bone marrow cells were obtained from rat femora. Cells cultured in the DMEM for one week at 37 °C were then cultured with nanocomposite at 100,000 cells/0.5 mL in each well. Then, half of the bone marrow cells were differentiated into osteoblast cells by adding dexamethasone, ascorbic acid, and beta-glycerophosphate in the following concentrations: dexamethasone 40 ng/mL (100 nM); ascorbic acid 17.6 μg/mL (100 μM), and beta-glycerophosphate 10 mM. Then, together with the nanocomposite, 20,000 raw cells/0.5 mL were added into each well. In addition, half of the raw cells were differentiated into osteoclasts by RANKL and mouse GM-CSF. The final concentration of RANKL was 50 ng/mL, and the final GM-CSF was 10 ng/mL. Raw and bone marrow cells were incubated at 37 °C for 14 days and 10 days, respectively, according to the specifications. Then, the cells were lysed using the cell lysis buffer, which was added in the amount of 400 µL to each well, incubated at 37 °C for 1 h, sonicated for 10 min, and transferred into a tube.

An ALP assay for bone marrow cells was performed as follows: 100 µL of ALP assay buffer, 100 µL of the sample, and 20 µL of substrate were added into each well and incubated for 40 min at 37 °C; then, 50 µL of NaOH (0.25 M) was added to stop the reaction. The optical density was read at 405, and the ALP/µg of total protein was calculated. The TRAP assay for osteoclast cells included 150 µL of assay buffer (pH 5.0–5.8) added to 50 µL of the sample and incubated at 37 °C for 30 min, and the OD was 405.

### 3.5. Mechanical Properties

Mechanical properties of the prepared composites were tested using the compression test machine of the 810 Material Testing System (universal testing machine, Eden Prairie, MN, USA) according to ASTM-D1621, with a loading rate of 1.27 mm/min compression, until the sample was compressed to 30% of its original size. The compression modulus and yield strength were calculated from the tested data. During the experimental studies, at least five experiments were conducted on the prepared samples, and the test results were averaged. Further explanation of these tests was provided in detail elsewhere [23].

## 4. Conclusions

A simple and cost-effective melt-casting and salt porogen leaching fabrication technique were used to fabricate PEEK bionanocomposites with HA and carbon nanoparticles (CNTs and CF). This fabrication technique is widely viable for highly porous foams since 85% porosity can be effectively achieved. About 20 wt% HA was chosen since it is a greater enhancer of bone growth. The micro-CT test confirmed that the fabrication technique used in this research is reliable because the PEEK foams exhibited a uniform pore size and good interconnectivity. Porosity variation compared to design porosity was 4%, except in the case of the 85% porosity nanocomposite, which varied by around 10%. In the biological tests, the nanocomposite proved to be non-toxic and had good cell viability, where a smaller addition of CNTs yielded better results. In the cell viability tests, PEEK/HA composites with 0.5 wt% CNTs established a good attachment compared to the neat PEEK foam. Additionally, HA enhances cell viability more than other inclusions. A similar performance was seen in culture tests of bone marrow cells (osteoblasts and osteoclasts). The 0.5% CF for osteoblasts and 1% CNTs for osteoclasts showed higher cell attachment and better performance. The test results have proven that PEEK/HA with CF and CNT bionanocomposites can be a potential candidate for bone scaffolding and other tissue engineering materials.

## Figures and Tables

**Figure 1 molecules-25-03572-f001:**
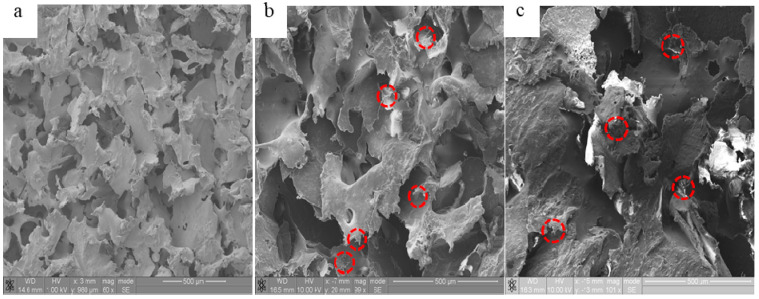
SEM micrographs of bionanocomposites with 75% porosity: (**a**) PEEK only, (**b**) PEEK/HA (20 wt%)/CNTs (0.5 wt%), and (**c**) PEEK/HA (20 wt%)/CF (0.5 wt%).

**Figure 2 molecules-25-03572-f002:**
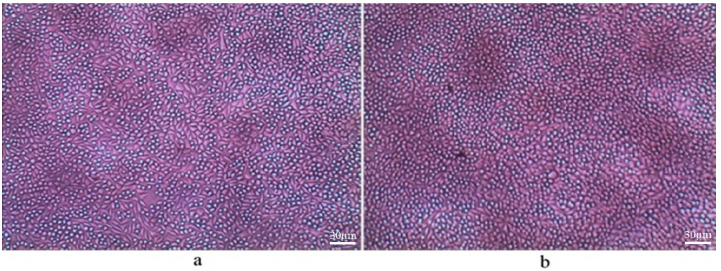
Day 3 cell culture of PEEK/20 wt% HA/ 0.5 wt% CF: (**a**) 75% porosity and (**b**) 85% porosity, concentration 1:4.

**Figure 3 molecules-25-03572-f003:**
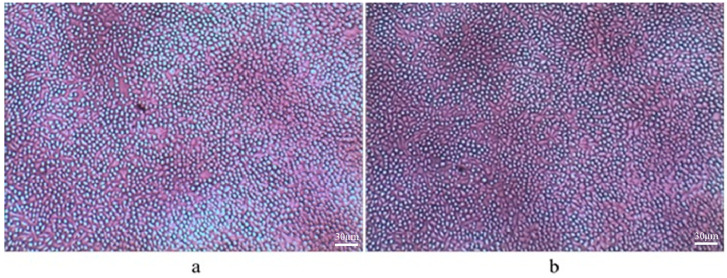
Day 3 cell culture of PEEK/20 wt% HA/1.0 wt% CNTs: (**a**) 75% porosity and (**b**) 85% porosity, concentration 1:4.

**Figure 4 molecules-25-03572-f004:**
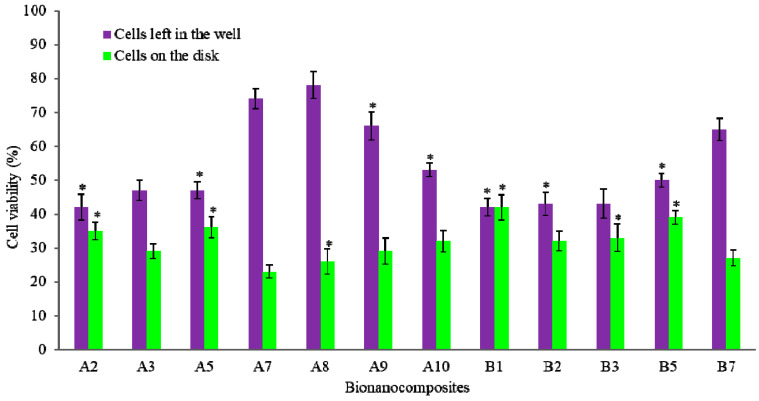
Cell viability results of PEEK and its nanocomposites. * represents *p* < 0.05.

**Figure 5 molecules-25-03572-f005:**
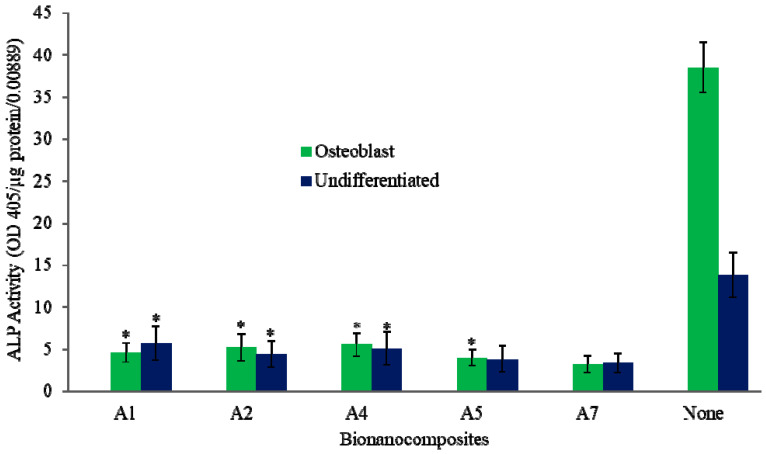
ALP for lysis of bone marrow cells in bionanocomposites with 75% porosity. * represents *p* < 0.05.

**Figure 6 molecules-25-03572-f006:**
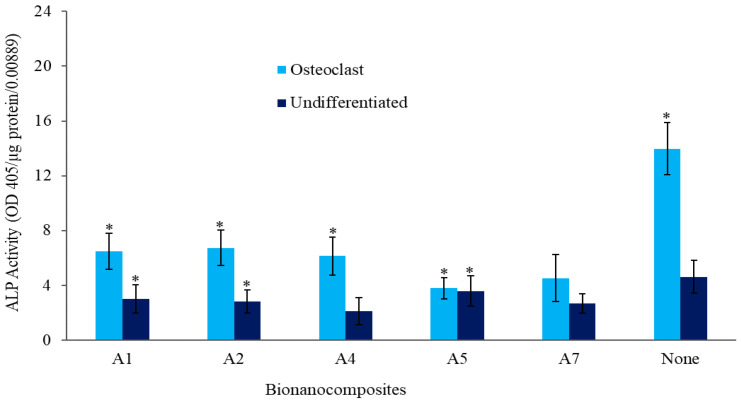
TRAP expression of raw cells in bionanocomposites with 75% porosity. * represents *p* < 0.05.

**Figure 7 molecules-25-03572-f007:**
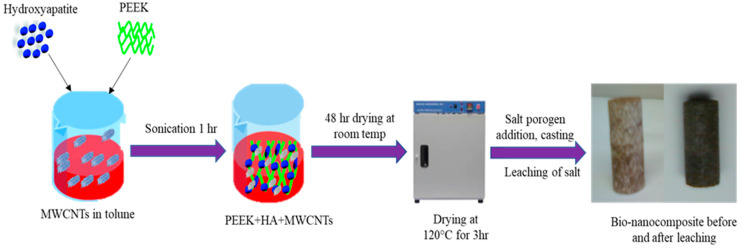
Schematic of PEEK/HA/CNT bionanocomposite synthesis process for biomedical applications.

**Table 1 molecules-25-03572-t001:** Bionanocomposite samples’ codes, compositions, and various tests performed in the study.

Sample Code	Sample Composition	Test Performed
A1	PEEK/HA (20 wt%)/CNT (0.5 wt%), porosity 75%	ALP, Mechanical
A2	PEEK/HA (20 wt%)/CNT (1.0 wt%), porosity 75%	ALP, Mechanical, Cell viability
A3	PEEK/HA (20 wt%)/CNTs (2 wt%), porosity 75%	Cell viability, Mechanical
A4	PEEK/HA (20 wt%)/CF (0.5 wt%), porosity 75%	ALP, Mechanical
A5	PEEK/HA (20 wt%)/CF (1.0 wt%), porosity 75%	ALP, Mechanical, Cell viability
A6	PEEK/HA (20 wt%) /CF (2 wt%), porosity 75%	Mechanical
A7	PEEK only, porosity 75%	ALP, Mechanical, cell viability,
A8	PEEK/HA (10 wt%), porosity 75%	Cell viability
A9	PEEK/HA (15 wt%), porosity 75%	Cell viability
A10	PEEK/HA (20 wt%), porosity 75%	Cell viability, Mechanical
B1	PEEK/HA (20 wt%)/CNTs (0.5 wt%), porosity 85%	Cell viability, Mechanical
B2	PEEK/HA (20 wt%)/CNTs (1 wt%), porosity 85%	Cell viability, Mechanical
B3	PEEK/HA (20 wt%)/CNTs (2 wt%), porosity 85%	cell viability, Mechanical
B4	PEEK/HA (20 wt%)/CF (0.5 wt%), porosity 85%	Mechanical
B5	PEEK/HA (20 wt%)/CF (1 wt%), porosity 85%	Cell viability, Mechanical
B6	PEEK/HA (20 wt%)/CF (2 wt%), porosity 85%	Mechanical
B7	PEEK only, porosity 85%	Cell viability

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
