# Peer review of "Fabrication and Biological Analysis of Highly Porous PEEK Bionanocomposites Incorporated with Carbon and Hydroxyapatite Nanoparticles for Biological Applications"

_molecules, 2020, doi:10.3390/molecules25163572_

Round 1

Reviewer 1 Report

The manuscript entitled ”Fabrication and biological analysis of highly porous PEEK bionanocomposites incorporated with carbon and hydroxyapatite nanoparticles for biological applications”.

In general, the manuscript is interesting and the goals set by the Authors are worth achieving.

However, the main problem with this work is that  it duplicates research previously published by the same researchers (Uddin et al. Progress in Biomaterials 2019). Mechanical properties described in the article “Mechanical properties of highly porous PEEK bionanocomposites incorporated with carbon and hydroxyapatite nanoparticles for scaffold applications” were presented in Figure 3 and Figure 4 while in present manuscript the data are collected in Table 1. The same problem is with CT results, which are described in the previous article: in present manuscript the Authors do not even refer to their published work here.

The biological tests: all the obtained samples did not study in the same manner. Why?

The way of the results description is a bit chaotic and should be improved.

Therefore I cannot recommend this paper to be published as it is in Molecules. 

Reviewer 2 Report

The paper discusses the biological, mechanical, and morphological characteristics of porous PEEK HA (CF or CNTs) scaffolds prepared by salt leaching method. The presentation of mechanical characteristics is described adequately, proving that the composites studied in the manuscript have mechanical properties which fall within the range of cancellous bone. I have some observations and recommendations for the other sections of the manuscript, which are described below:

  1. In the introduction (line 94-95) the authors state that highly robust PEEK scaffolds incorporated with hydroxyapatite and carbon particles were produced here for the first time. However, as the current manuscript is built on a previous study (Ref. 25) , I suggest to the authors to describe the relation between that article and the current manuscript and to reference the article in this section.
  2. The section 2.2. from Results & Discussions describe the micro-CT which were obtained during the previous study, without providing a reference (lines 128-136). Moreover, the SEM results are described only briefly in the text. I would suggest to present the CT results more concisely and with a reference, and to discuss the SEM results in more detail.
  3. The biological analysis sections were greatly improved, as compared to previous versions of the manuscript, as well as the figures 2,3,4,5 and 6. However, it is still necessary to provide p-values for the presented results (Fig.4, Fig.5, and Fig.6).
  4. Other minor comments:

4.1) Please remove the last phrase from lines 108-109 regarding the mechanical testing performed in triplicate. This information is already presented in the Materials & Methods section.

4.2) Please remove the melting point of HA from line 222. Although the information was probably presented by the manufacturer, HA is known to be thermally stable for higher temperatures and its melting point is not discussed in the literature (for more details, check DOI:10.1007/s10973-011-1877-y).

4.3) Please rename section 3.3. “Morphology”, “Microscopic Analysis” or similar term, since micro-CT test was removed from this study. Also, please remove lines 283-297 related to the test from the manuscript.

Round 2

Reviewer 1 Report

The Authors corrected the text according to the suggestions. However, the main problem of this article was not resolved: in my opinion some of results duplicate research previously published by the same researchers (Uddin et al. Progress in Biomaterials 2019). 

Author Response

Thank you for the reviews/comments. This work was a part of the Ph.D. studies, and the design and fabrication part was published in Uddin et al. Progress in Biomaterials 2019, while the biological part was submitted to this journal. Only a small portion was added in this part of the manuscript based on the reviewer’s comments".

Thanks in advance

Md. Nizam Uddin